# Examining Drinking Water Preferences among University Students: A Comparative Assessment

**DOI:** 10.3390/ijerph21101271

**Published:** 2024-09-25

**Authors:** Razi Mahmood, Norr Hassan, Ashraf Chamseddine, Ravi Rangarajan, Rami Yassoub

**Affiliations:** 1Department of Public Health, AFG College with the University of Aberdeen, Doha P.O. Box 10805, Qatar; razi.mahmood@afg-aberdeen.edu.qa; 2Department of Environmental Health and Safety, University of Doha for Science and Technology, Doha P.O. Box 24449, Qatar; ashraf.chamseddine@udst.edu.qa (A.C.); ravi.rangarajan@udst.edu.qa (R.R.); 3Department of Quality, Patient Experience and Health Informatics, Sidra Medicine, Doha P.O. Box 26999, Qatar; ryassoub@sidra.org

**Keywords:** bottled water, tap water, drinking water, university students, preferences, behavior, cross-sectional study

## Abstract

In recent years, there has been a clear increase in bottled water consumption globally, which has led to significant health and environmental concerns. This cross-sectional study aims to understand the attitude and preferences of university students in Qatar towards the use of bottled and tap drinking water using an online survey questionnaire (September and October 2022). The number of students who responded to the online survey was 14% (*n* = 688) of the student population, with a mean age of 22.23 ± 5.15 years from the different colleges. Overall, a higher fraction of students preferred plastic bottled water as the main drinking source on campus (*n* = 468; 68.02%), with a majority of them being females (72.08%). Out of the 468 students who preferred plastic bottled water, safety was the most important factor (43.80%), followed by convenience (16.88%) and taste (15.60%). Cost (15.17%), personal/family habits (5.13%), environmental concerns (2.14%), and mineral content (1.28%) were found to be the least important factors. Among the 45 students (6.54%) who preferred tap water over plastic bottled water, cost (46.67%) was the dominant factor, followed by convenience (20.00%), environmental concerns and safety (13.33% each), taste (4.44%), and personal/family habits (2.22%). Around 72% believed that plastic water bottles were more harmful to the environment, yet the greater majority still resorted to this source. The research study underscores safety as the major factor favoring bottled water over tap water. Further, it suggests that knowledge alone does not fully explain student behavior, implying other influential factors. This study recommends campaigns focus on attitude and behavior change and not solely emphasize knowledge. There is an immediate need to further educate students through environmental and health literacy programs on water consumption and quality. Enabling the population to understand the positive and negative aspects of their choices may be an effective remedy for ensuring a healthy population and healthy environment.

## 1. Introduction

The provision of safe and easily accessible water plays a crucial role in safeguarding public health [1]. Many countries have implemented measures to ensure the supply of clean and safe drinking water via tap water systems that meet basic drinking standards [2]. Nevertheless, in recent decades, there has been a noticeable increase in the global consumption of bottled water, particularly in developed countries, despite the fact that tap water is safe for direct consumption [3]. Bottled water consumption is mainly attributed to factors like safety and health, taste, convenience, marketing features, and lifestyle choices [4,5,6,7,8,9,10]. The perception that bottled water is a healthier and safer option than other water sources like tap water has significantly impacted consumption patterns [4,5,6,10,11,12,13,14,15]. However, there exists a source of bias centered around people’s perceptions impacting these consumption trends [16,17], and the factual basis of this perception should be understood to make informed choices about water consumption at community and national levels, with a keen focus on their public and environmental health implications. 

In most high-income countries, the provision of tap water is highly regulated, and quality parameters are publicly disclosed to spread awareness about the safety of tap water [13,14,18,19]. On the other hand, in most low- and middle-income countries (LMIC), these water quality and safety specifications are not shared, leading to their populations, especially the younger generations, resorting to bottled water as a more trusted source [17,20,21]. 

As for the affluent yet naturally water-scarce Gulf Cooperation Council (GCC) countries, heavy reliance on seawater desalination exists [22]. Organoleptic parameters and physiochemical properties in the desalinated sea water have led to an increased dependency on plastic bottled water consumption in the region [22,23]. 

In the State of Qatar, the Qatar General Water and Electricity Corporation, known as KAHRAMAA, provides drinking water through the desalination of seawater. KAHRAMAA developed its own drinking water quality requirements and conditions in line with international World Health Organization (WHO) guidelines, ensuring the availability of safe and organoleptic-pleasing water. KAHRAMAA collects more than 12,000 water samples from approximately 700 sampling points annually, providing confidence in the safety and drinkability of municipal tap water [24]. According to a study conducted by the Qatar Environment and Energy Research Institute (QEERI) in 2015, a total of 113 tap water samples and 62 bottled water samples were collected and tested. The findings revealed that tap water in Qatar meets the standards established by both the WHO and the US Environmental Protection Agency (EPA) [25]. However, despite the availability and assurances of safe tap water, there is still a significant consumption of bottled water in the country. Data obtained from interviews conducted by KAHRAMAA in people’s homes indicate that only 30% of Qatar residents consume tap water [26]. It is evident that there is no substantial advocacy in Qatar to encourage and motivate people to consume tap water, despite the comprehensive measures taken to ensure the safety and quality of municipal tap water. 

In relation to extensive bottled water consumption regionally and globally, environmental concerns arise, ranging from high energy usage, increased carbon emissions during production and distribution, plastic pollution, and the depletion of groundwater resources [16,27]. In 2019, the global emissions resulting from the life cycle of all plastics amounted to 860 million tons of carbon dioxide (CO_2_), a figure comparable to CO_2_ released by 189 coal plants [28]. Every minute, over 1,000,000 bottles are sold globally [29], with most of them being single-use bottles and about 85% of all plastic bottles sold becoming waste. The disposal of these inadvertently leads to excessive plastic pollution [30]. The predominant packaging material used for bottled water is polyethylene terephthalate (PET), derived from fossil fuels [16], and can persist for at least 450 to 1000 years before it fully decomposes [31], thus overloading landfill space for an extended period of time [32]. 

Understanding the implications of bottled water consumption necessitates investigating consumer preferences and attitudes. University students represent a significant demographic sector and future consumer sector of the community, especially in Qatar. Hence, understanding their attitude and preferences towards the use of plastic bottled water is essential. Few studies have delved into the drinking water choices of university students, shedding light on factors influencing their preferences, including safety perceptions, convenience, taste, and even cultural habits [3,16]. The study exploring drinking water choices at Purdue University revealed that women drink disproportionately more bottled water than men, while undergraduate students drink more than graduate students, staff, and faculty. Important barriers that were established in this study include perceived risks from tap water, perceived safety of bottled water, preferred taste, and convenience of bottled water [3]. In another cross-regional comparative study conducted among East Asian university students across Singapore, Hong Kong, and Macau, it was noted that Singapore has a relatively low rate of bottled water consumption among its university students, while in Hong Kong and Macau, one-fourth of the students still drink bottled water more frequently than tap water. In terms of determinants of drinking water choices, “safety and hygiene” and “convenience and availability” ranked the highest for all three regions compared. However, factors like “taste”, “price”, and “personal and family habits” were less preferred by different sub-samples within the study population [16].

Currently, little is known about the west Asian region (Middle East), and there are no published studies on Qatar that address the attitude and preferences of drinking water choices among students on university campuses. Exploring these preferences helps in devising informed and evidence-based strategies related to public policy, behavioral interventions, and sustainable efforts to encourage sustainable and mindful drinking habits within academic communities. Thus, a study has been conducted within a public university in Qatar (University of Doha for Science and Technology, Qatar) to address the following research questions:What are the current attitudes toward drinking water choices on university campuses?What are the major factors that determine the drinking water choices of university students?

## 2. Materials and Methods

### 2.1. Study Design and Participants

This cross-sectional study utilized a structured survey administered to university students to gain insights into their attitudes and preferences regarding drinking water. This approach allowed for a comprehensive assessment of patterns and correlations within a large and diverse sample of participants, establishing statistical relationships between the variables considered. The study was administered to all full-time students registered at the University of Doha for Science and Technology (N = 4920), covering all the colleges (Business Management, Computing and IT, Engineering Technology, Health Sciences, and the Foundation program). The survey and reminders (sent two weeks after the original survey distribution to all students) were sent between September and October 2022 via the university’s online survey software, i.e., Blue software 8.0 from Explorance. The invitation included a study description and letter of consent, emphasizing the anonymous and voluntary nature of participation with no penalties for refusal to participate.

### 2.2. Framework

Several theoretical frameworks have been developed to explain the factors that determine certain behaviors. In this study, concepts from previous studies were applied by incorporating the personal norms and behaviors that are predicted by behavioral intentions into the theory of planned behavior (TPB). According to TPB theory, behaviors are predicted by personal intentions (including cost, taste, safety, minerals, environmental concerns, family habits, and convenience), which are preceded by personal attitude, perception of norms, perceived behavioral control, and self-expectations [3,33,34,35]. An individual’s perception of norms refers to how others would view them, particularly pro-environmental behavior, and the motivation to comply with their views. Moreover, perceived behavior control relates to individuals’ perception of the extent to which the performance of the behavior is easy or difficult. It is noteworthy to mention that in this study, drinking tap water versus bottled water is considered to be a personal intention-affected behavior. For this reason, a framework showing the factors that impact different behaviors of drinking water choices has been used to develop the survey and is depicted in Figure 1. According to this proposed framework, the personal attitude toward drinking water choices will be affected by gender, age, nationality, college to which students belong, number of years spent at university, hours per week spent on campus, average daily water consumption on campus, and major source of water used on campus, which in turn contribute heavily to university students’ personal intentions. This theoretical application allows for a structured examination of preferences and perceived behavioral control in influencing water consumption choices.

### 2.3. Survey Questionnaire Development and Pilot Testing

The survey was developed from peer-reviewed articles addressing relevant subjects and outlining factors influencing preferences in water consumption among university students, including Qian (2018) [3] and Abdah et al. (2020) [36]. These articles served as the foundation for constructing the questionnaire, and the questions were contextualized and reframed based on the well-established framework of the theory of planned behavior given in Figure 1 [3,33,34,35]. There was a rigorous process of pilot testing, and a pilot survey was distributed to a representative group consisting of three university faculty members, one research assistant, and three undergraduate students for feedback on the clarity, relevance, and layout of the questionnaire. Based on the feedback received, revisions were made, and the group was re-consulted prior to finalizing the questionnaire. The results of the pilot testing were not used in the analysis. The finalized study questionnaire consisted of 15 items covering demographic factors such as gender, age, nationality, college, years at university, and hours per week spent on campus. Additionally, it examined attitudes and preferences regarding plastic bottled water compared to tap water, encompassing daily consumption on campus, major drinking water sources used, convenience, safety concerns, taste, environmental concerns, mineral content, cost, and personal/family habits.

### 2.4. Variables

The outcome variable for this study was the preferred water source on campus. This categorical variable had the following responses: plastic bottled water, tap water, and no preference. The independent variables for this study included one continuous factor that is age (years), while the rest were categorical, including the following: gender (male or female), nationality (re-coded to Qatari or non-Qatari), college unit (business management, computing and IT, engineering technology, health sciences, and foundation program), number of years at university (up to 1 year; from 1 to 2 years; from 2 to 3 years; from 3 to 4 years; or more than 4 years), hours per week spent on campus (less than 10 h; from 10 to 20 h; from 21 to 40 h; or more than 40 h), and average daily water consumption on campus (less than 200 mL; from 200 to 330 mL; from 330 to 500 mL; from 500 mL to 1 L; or more than 1 L). Students were further surveyed about which source of water (categorical, plastic bottled water, tap water, no difference, or do not know) was more convenient, safer, tasted better, had more mineral content, and posed a higher environmental concern. Finally, the participants were requested to rank their water preferences by assigning a score of 1 (most important) to 7 (least important) on the following factors: cost, convenience, safety, taste, environmental concerns, minerals, and personal/family habits. 

### 2.5. Statistical Methods

Descriptive statistics of the sample were presented as mean and standard deviation for continuous variables and frequency and proportion for categorical variables. These results were stratified by preferred water source on campus (Group 1: plastic bottled water; Group 2: tap water; or Group 3: no preference). To examine whether differences exist between the three groups, a one-way ANOVA or the Kruskal–Wallis nonparametric test (if normality or homogeneity of variance assumptions were not met) was used for continuous variables (age) due to a violation of the assumption of normality. The chi-squared test was used to test if there was a statistically significant association between categorical variables. In cases where the expected count in the cells for two categorical variables was less than five in more than 20% of cells, Fisher’s exact test was used. Similarly, the chi-squared test or Fisher exact test was used when assessing the knowledge, attitudes, and beliefs of students stratified by their preferred water source. Cramer’s V was used to assess the strength of association between the variables (0–0.05 no or very weak association; 0.05–0.10 weak association; 0.10–0.15 moderate association; 0.15–0.25 strong association; and >0.25 very strong association [37,38]. When examining the ranking of factors influencing water preference on campus ordinal data with a ranking of 1 (most important) to 7 (least important), proportions, median (interquartile range), and stratification by plastic, bottled, or tap water preference were reported for each of the 7 individual factors. The Kruskal–Wallis test was used to determine if any statistically significant differences in the distribution of ranking by preferred water source (plastic bottled water or tap water) existed. A 5% level of significance was used for all analyses which were conducted using The Statistical Package for Social Sciences (SPSS), version 28.0.10.

### 2.6. Ethical Considerations

The study proposal was reviewed and granted exemption under research category (2) from the Health Research Governance Department at the Ministry of Public Health (MoPH) of Qatar (Approval Ref # ERC-61-2-2022). The data collected did not include any personal identifiers, and steps were taken to ensure confidentiality of the data, including securing data files with the Blue Explorance software (Version 1) program.

## 3. Results

### 3.1. Sample Characteristics 

In total, the study achieved a 14% (*n* = 688) response rate. The mean age of the respondents was 22.23 ± 5.15 years, where the majority were females (*n* = 437; 63.52%) and non-Qatari (*n* = 598; 86.92%). A large proportion were enrolled in the College of Health Sciences (*n* = 220; 31.98%), attending the university for one to two years (*n* = 181; 26.31%), and spending 21–40 h per week on campus (*n* = 254; 36.92%).

Overall, students reported plastic bottled water as the preferred source of drinking water (*n* = 468; 68.02%), followed by those with no specific preference (*n* = 175; 25.44%), and lastly, 6.54% preferred tap water (*n* = 45; 6.54%). There were statistically significant differences in preferences based on the average daily water consumption on campus (χ(8) = 19.64, *p*-value = 0.012, Cramer’s V = 0.169) and the main source of water used on campus (χ(6) = 120.13, *p*-value < 0.001, Cramer’s V = 0.295). Moreover, there was a higher proportion of females (72.08%) compared to males (60.96%) who preferred plastic bottled water, which was statistically significant (χ(2) = 10.19, *p*-value = 0.006, Cramer’s V = 0.122).

When examining additional factors and sub-groups, the following suggested preferences for plastic bottled water were identified: Qatari students (75.56%), Foundation Program Unit students (78.95%), students enrolled at the university for one to two years (76.24%), students who were on campus less than 10 h per week (73.68%), and students that drank between 200 and 330 mL of water daily on campus (76.09%). There were no statistically significant differences in preferred water source on campus by age, nationality, college, number of years at the university, or hours per week spent on campus (*p*-value > 0.05). In addition, there were observed discrepancies between the preferred choices indicated by respondents and their actual drinking water source on campus. For instance, 21.64% of students using plastic bottled water reported no preference in water source; however, 21.13% of tap water users and 38.10% of mixed users preferred plastic bottled water on campus, respectively.

Table 1 summarizes the study sample characteristics stratified by preferred source of water used on campus.

### 3.2. Knowledge, Attitudes, and Beliefs of Students about Water Sources on Campus

Results related to knowledge, attitudes, and beliefs indicated that the majority of respondents consistently selected plastic bottled water as more convenient (60.50%), safer (71.50%), and having a more acceptable taste (67.20%) compared to tap water. In addition, 71.7% of students indicated more environmental concerns related to plastic bottled water, while 50.4% of the students perceived bottled water to be richer in minerals. There was a statistical association between convenience (χ(6) = 18.07, *p* = 0.006, Cramer’s V = 0.188), safety (*p*-value < 0.001, Fisher’s exact test, Cramer’s V = 0.411), taste (*p*-value = 0.023, Fisher’s exact test, Cramer’s V = 0.167), environmental concerns (*p*-value < 0.001, Fisher’s exact test, Cramer’s V = 0.246), and preferred water source. No statistically significant correlation was found between mineral content and their preferred source of water.

Table 2 summarizes the knowledge and attitudes of students about sources of water on campus.

### 3.3. Ranking Factors Influencing Preferred Water Sources

When ranking factors influencing preference for water source, among 468 students who preferred plastic bottled water rather than tap water, safety was the most important factor (43.80%), followed by convenience (16.88%) and taste (15.60%). Cost (26.50%) and personal/family habits (24.57%) were found to be the least important of the factors listed for our study respondents. Interestingly, of the 45 students who preferred tap water over plastic bottled water, cost was identified by 46.67% of respondents as the most important factor influencing their preference, followed by convenience (20.00%). Personal/family habits were most commonly reported as the least important factor influencing preference (44.44%), even among the tap water preferred cohort.

Table 3 shows the ranked factors influencing preferences for plastic bottled water (N = 468) or tap water (N = 45).

Using the Kruskal–Wallis test, there were statistically significant differences in the distribution of the rankings of cost (χ^2^(1) = 13.97, *p*-value < 0.001), convenience (χ^2^(1) = 7.41, *p*-value = 0.006), safety (χ^2^(1) = 35.52, *p*-value < 0.001), taste (χ^2^(1) = 10.63, *p*-value = 0.001), environmental concerns (χ^2^(1) = 12.53, *p*-value <0.001), and personal/family habits (χ^2^(1) = 4.84, *p*-value = 0.028) when comparing individuals who preferred plastic bottled water to those who preferred tap water on campus. No statistically significant differences in the distribution of the ranking of mineral content between the two groups (*p*-value > 0.05) were found. Cost was identified as having the most importance to students selecting tap water (median = 2.0, IQR = 4.5), while safety had the highest median score among plastic bottled water users (median = 2.0, IQR = 2.0).

Table 4 provides the median, interquartile range (IQR), and *p*-value for within factor differences in the distribution of rankings.

## 4. Discussion

The results of this study provide a baseline for future research into the water preferences of students in Qatar. The majority of the respondents indicated a preference for plastic bottled water on campus, with a higher proportion of females. Plastic bottled water was identified as being safer, more convenient, and tasting better. There were gaps identified in knowledge pertaining to environmental concerns and mineral content by source of water. Safety and taste were the highest-ranked drivers of plastic bottled water preferences, while cost and convenience were drivers for students who preferred tap water.

### 4.1. Preferred Water Source on Campus

A study conducted in the United Arab Emirates found approximately 80% of the drinking water supply for persons aged up to 18 years was through bottled water [39]. Another study conducted in the West Bank found 92% of students drank bottled water on campus [36]. One study from Spain indicates that roughly only 50% of respondents use bottled water as their primary drinking water source [40], and even lower rates (ranging between 9.68% and 19.05%) for daily bottled water use on campus were recorded amongst individuals from Singapore, Hong Kong, and Macau [3]. In Mexico, individuals on campus preferred to buy bottled water even when the quality of the tap water system was met [41]. This could be due to the lack of trust in and overall accessibility of tap water from municipalities, which encourage citizens to depend on plastic bottled water [42]. Here, we argue the need to study the trend that developed countries, with substantial efforts in investing in public water networks and higher per capita, give less preference to bottled water due to their increased confidence in their water systems. This may allow the hypothesis that, without cost being a factor, human behavior tends to choose the safest option.

### 4.2. Gender Differences in Water Source Preferences

In this study, when examining preferences by gender, a higher proportion of females indicated a preference for plastic bottled water compared to males. This is a consistent trend supporting the existence of gender differences in water preferences, with a previous study reporting that females tend to consume more bottled water than males [16] and females having a higher risk perception towards tap water [43]. A study conducted in the United States showed that females have 1.32 times higher odds of consuming bottled water, even when other factors like level of education, etc., are equal compared to males [7]. In addition, it has also been established that gender differences influence the ranking of tastes and personal/family habits [3,36]. There are two opposing effects hypothesized to influence gender-based preferences. On one side, females have negative perceptions of the safety of tap water [19,43], while at the same time, females may be more environmentally conscious [19]. Thus, further investigation is required to determine what the factors driving choices among males and females in water preferences are.

### 4.3. Average Fluid Consumption on Campus

The average daily consumption of drinking water on campus, regardless of its source, varied among students. Sixty-one students (8.86%) reported drinking less than 200 mL per day on campus, which is well below the recommended 2600 mL of daily fluid consumption to maintain water balance in the body of an adult individual [44,45,46]. Our study showed a statistically significant decrease in plastic bottled water consumption with the overall increase in average daily water consumption (from those drinking <200 mL, 72.13% chose plastic bottled water, compared to 55.79% who drank >1 L per day). This could be attributed to students who are more health-conscious (in that they ensure to drink sufficient amounts of water per day) and who are also not necessarily just more aware of the dangers of plastic products but have researched enough to be confident in tap water’s chemical and biological properties.

### 4.4. Health Concerns

Contaminated tap water has been associated with gastrointestinal illnesses, developmental delays, cognitive impairment, cardiovascular disease, cancer, fertility and birth complications, and even death, which are all prominent health concerns linked to unsafe drinking water [47,48,49,50]. In this study, 43.80% of the students prioritized safety over all other factors when it comes to choosing plastic bottled water over tap water. This emphasis on safety could be attributed to the prevalent news coverage regarding contamination issues associated with tap water. Consequently, it is expected that people would necessarily resort to an alternative like bottled water. Although bottled water has attracted a health-seeking cohort of people wishing to escape the fast-urbanizing and inadequately maintained public water networks of overflowing cities, they too have presented their own health issues and substantial environmental impacts throughout their lifecycle. Spanning from their method of manufacture to their storage and exposure to the natural elements (the presence of micro-nanoplastics due to the mechanical stress on the plastic products) [51], and up to their type of disposal, plastic bottled water’s negative impacts on human health are numerous, including the prevalence of endocrine-disrupting chemicals, metabolic disorders, tissue necrosis, blood vessel embolization, apoptosis, and carcinogenicity [52,53,54,55,56,57,58,59]. While it is very important to acknowledge the health drawbacks of contaminated tap water, it is equally crucial to highlight the various health effects of plastic bottled water from production to disposal, which unfortunately tends to receive less discussion.

### 4.5. Environmental Concerns

Plastic production processes proved to be more harmful to the environment, especially in the context of energy-intensive technologies used, compared to drinking water treatments [60]. Also, only 14% of the 78 million metric tons of plastic packaging produced annually becomes recycled, and the remaining ~80% finds its way into the oceans, leading to the well-documented marine plastic pollution [61,62]. Interestingly, in our study, environmental concerns were not highly ranked among factors influencing water preferences on campus, although they were higher among those who chose tap water compared to plastic bottled water. That comes with 71.7% of students indicating that plastic bottled water has a more negative environmental impact compared to tap water, while 8.6% said that tap water poses a greater environmental threat. This figure is exacerbated by another 4.8% that claim there is no difference at all between the two sources and 15% that do not know if the environmental impact exists.

With the observed global trend of ~30% of the students being unaware of environmental concerns like plastics contribution to global warming, plastic pollution, including microplastics and nanoplastics in food and water sources [63,64], and the ever-increasing anthropogenic impacts on the environment, community-level environmental health literacy is of paramount importance among the student community, especially.

### 4.6. Study Implications

Our current study evaluated student behavior within the framework of the Theory of Planned Behavior and demonstrated that the key factors, seen across the world, that have promoted the favoring of plastic bottled water over tap water apply to our study region too. The uniqueness of our study rests in its focus on understanding the preferences of university students in a high-income GCC country like Qatar. It reinstates the importance of aspects such as taste and convenience, but of particular interest is its manifestation that knowledge, or lack thereof, is not the sole culprit in the rise and prevalence of plastic bottled water dependency. Although almost 15.00% of the respondents within this higher education cohort assert no to the weak unknown correlation between plastic bottled water and its detrimental impact on the environment, it is the relative importance they give to safety, more than any other factor, that should be considered as the guiding compass for future public health policies and campaigns related to plastic bottled water use. It is arguable that academic and research institutions as well as non-governmental environmental organizations may have wielded a double-edged sword in their eagerness to address health concerns with increased emphasis on lack of public water network maintenance, leading to people opting for the alternative plastic bottled water, a detrimental source rooted in deep environmental concerns [65]. Accordingly, to reverse this trend, policymakers and environmental advocates need to acknowledge and act upon the fact that people value their immediate health and safety (especially in the short term) over the environmental implications of their actions [66,67]. Therefore, complementary efforts highlighting the consequences of plastic bottled water use with concurrent continued investment in improving public water networks in countries like the State of Qatar (where tap water meets safe drinking standards) may increase and improve the public’s awareness of this issue.

Additionally, campaigns to deter negative health and environmental practices need to focus on attitude and behavior change and not solely emphasize knowledge; 71.7% of our respondents believed that plastic water bottles are more harmful to the environment, and yet the greater majority still resorted to this source.

Our knowledge to practice recommendations is for institutions to test and publish results indicating the safety of the chemical and biological properties of tap water sources regularly to attract the health conscious among the student community within university settings. If tests reveal suboptimal properties, deliberations with the university board and ministries of education, environment, and energy should be sponsored to install the fixes needed to ensure safe tap water. The success of these efforts should be distributed annually to all academic institutions, both public and private, in the country. Data safety sheets available at public and tap water sources may serve as a powerful reassurance factor and need to be studied. This recommendation is supported by a study indicating a significantly greater proportion of GCC residents view tap water quality as poor (36.6%) compared to North American and European residents (23.8%) [68]. 

We advocate that enabling the population to understand the positive and negative aspects of their choices towards bottled water or tap water use and improving their ability to easily and accurately access information about alternative health and environmentally friendly sources and practices may prove to be the most cost-effective measure from an environmental health standpoint. 

### 4.7. Limitations

This study on the drinking water preferences of university students has some limitations that should be recognized and addressed in future research. Primary, the factor that may limit the generalizability of the findings is the sample size of university students (688 students; 14% response rate), which may not represent the larger population in terms of nationalities, cultural backgrounds, socioeconomic statuses, and more. Previous studies indicated that college students are more responsive to web surveys [69], but the relatively low response rate may influence results due to sampling bias. Another limitation of this study was the small sample size of students who preferred tap water on campus. This may limit the generalizability of the findings; however, these findings provide new insights into trends in tap water preferences among university students in Qatar, an area with a dearth of information. Finally, the study was conducted at one university campus, which is one of the largest campuses in Qatar. Thus, we recommend future studies study multiple public and governmental universities to enhance the robustness and applicability of the findings.

## 5. Conclusions

In conclusion, the results of this study found that the majority of students preferred plastic bottled water as their source of water on campus, with a small minority preferring tap water. A greater proportion of females reported a preference for plastic bottled water, indicating gender differences may exist, and factors such as safety and taste were the main drivers of plastic bottled water use. Furthermore, there was a significant proportion of students who were unaware of the environmental consequences of plastic bottled water use and the mineral content of the source of water.

## Figures and Tables

**Figure 1 ijerph-21-01271-f001:**
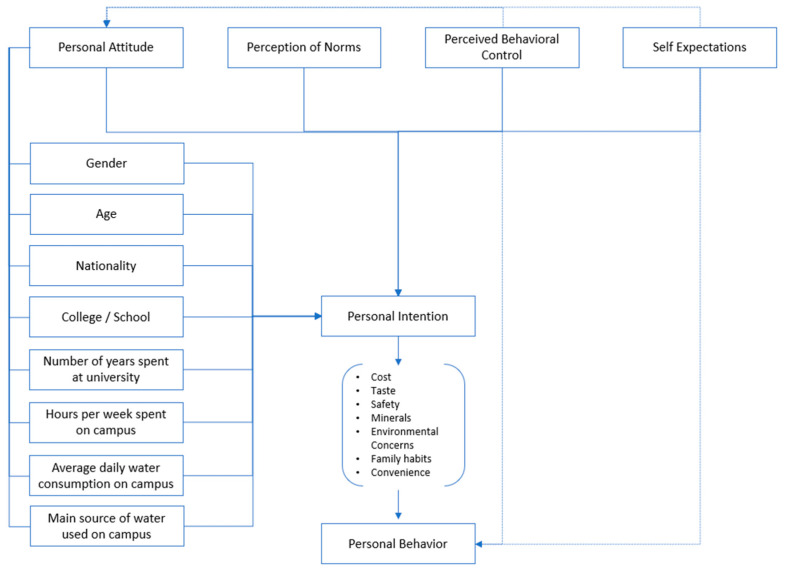
Proposed framework of the theory of planned behavior (TPB) [3,34,35].

**Table 1 ijerph-21-01271-t001:** Summary of sample characteristics (N = 688).

	Preferred Water Source Used on Campus			Total	*p*-Value χ^2^, (V)
	Plastic Bottled Water (*n* = 468; 68%)	Tap Water (*n* = 45; 6.5%)	No Preference (*n* = 175; 25.4%)	N = 688	
Gender					
Male	153 (60.96%)	17 (6.77%)	81 (32.27%)	251 (36.5%)	0.006 * 10.19, (0.122)
Female	315 (72.08%)	28 (6.41%)	94 (21.51%)	437 (63.52%)	
Age (Years ± SD)	23.21 ± 5.26	23.73 ± 5.67	23.30 ± 4.70	686 (100%)	NS
Nationality					
Qatari	68 (75.56%)	2 (2.22%)	20 (22.22%)	90 (13.1%)	NS
Non-Qatari	400 (66.89%)	43 (7.19%)	155 (25.92%)	598 (86.9%)	
College					
Business	148 (75.13%)	10 (5.08%)	39 (19.80%)	197 (28.6%)	NS
Computing and IT	75 (69.44%)	6 (5.56%)	27 (25.00%)	108 (15.7%)	
Engineering Tech	91 (63.19%)	8 (5.56%)	45 (31.25%)	144 (20.9%)	
Health Sciences	139 (63.18%)	21 (9.55%)	60 (27.27%)	220 (32.0%)	
Foundation	15 (78.95%)	0 (0.00%)	4 (21.05%)	19 (2.8%)	
Number of years at the university					
Up to 1 year	39 (58.21%)	6 (8.96%)	22 (32.84%)	67 (9.7%)	NS
1–2 years	138 (76.24%)	9 (4.97%)	34 (18.78%)	181 (26.3%)	
2–3 years	109 (66.46%)	12 (7.32%)	43 (26.22%)	164 (23.8%)	
3–4 years	119 (69.59%)	11 (6.43%)	41 (23.98%)	171 (24.9%)	
More than 4 years	63 (60.00%)	7 (6.67%)	35 (33.33%)	105 (15.3%)	
Hours per week spent on the campus					
<10 h/week	98 (73.68%)	6 (4.51%)	29 (21.80%)	133 (19.3%)	NS
10–20 h/week	163 (65.46%)	15 (6.02%)	71 (28.51%)	249 (36.2%)	
21–40 h/week	172 (67.72%)	19 (7.48%)	63 (24.80%)	254 (36.9%)	
>40 h/week	35 (67.31%)	5 (9.62%)	12 (23.08%)	52 (7.6%)	
Campus daily water consumption					
<200 mL	44 (72.13%)	3 (4.92%)	14 (22.95%)	61 (8.9%)	0.012 * 19.64, (0.119)
200–330 mL	70 (76.09%)	1 (1.09%)	21 (22.83%)	92 (13.4%)	
330–500 mL	151 (69.91%)	21 (9.72%)	44 (20.37%)	216 (31.4%)	
500 mL	150 (66.96%)	11 (4.91%)	63 (28.13%)	224 (32.6%)	
>1 L	53 (55.79%)	9 (9.47%)	33 (34.74%)	95 (13.8%)	
Campus main water source used					
Plastic Bottled Water	445 (74.66%)	22 (3.69%)	129 (21.64%)	596 (86.6%)	<0.001 * 120.13, (0.295)
Tap Water	15 (21.13%)	21 (29.58%)	35 (49.30%)	71 (10.3)	
Both Water Sources	8 (38.10%)	2 (9.52%)	11 (52.38%)	21 (3.1%)	

Note: Age (a continuous variable) is presented as mean +/− standard deviation, while the other variables (categorical) are presented as frequency (percentage). A *p*-value < 0.05 indicates significance. NS: not significant; (V): Cramer’s V statistic; * denotes the chi-square test.

**Table 2 ijerph-21-01271-t002:** Knowledge and attitudes of students about sources of water on campus (N = 688).

Factors	Plastic Bottled Water	Tap Water	No Difference	Do Not Know	*p*-Value χ^2^/FET, (V)
Convenience	416 (60.5%)	102 (14.8%)	170 (24.7%)	N/A	0.006 * 18.07, (0.188)
Safety	492 (71.5%)	30 (4.4%)	100 (14.5%)	66 (9.6%)	<0.001 FET, (0.411)
Taste	462 (67.2%)	30 (4.4%)	138 (20.1%)	58 (8.4%)	<0.001 FET, (0.167)
Environmental Concerns	493 (71.7%)	59 (8.6%)	33 (4.8%)	103 (15.0%)	0.005 FET, (0.246)
Mineral Content	347 (50.4%)	101 (14.7%)	69 (10.0%)	171 (24.9%)	NS

Note: A *p*-value < 0.05 indicates significance; NS: not significant; FET: Fischer’s exact test; (V): Cramer’s V statistic; * denotes the chi-square test.

**Table 3 ijerph-21-01271-t003:** Ranked factors influencing preferences for plastic bottled water (N = 468) or tap water (N = 45).

Factors	1 Most Important	2	3	4	5	6	7 Least Important
Cost							
Bottled	15.17%	15.60%	11.11%	11.11%	6.84%	13.68%	26.50%
Tap	46.67%	15.56%	4.44%	2.22%	6.67%	6.67%	17.78%
Convenience							
Bottled	16.88%	12.18%	11.75%	9.62%	14.10%	20.90%	15.38%
Tap	20.00%	28.89%	17.78%	6.67%	2.22%	20.00%	4.44%
Safety							
Bottled	43.80%	19.66%	19.44%	8.33%	4.91%	1.28%	2.56%
Tap	13.33%	4.44%	15.56%	37.78%	13.33%	15.56%	0.00%
Taste							
Bottled	15.60%	28.42%	21.37%	12.82%	8.55%	8.97%	4.27%
Tap	4.44%	17.78%	26.67%	11.11%	13.33%	13.33%	13.33%
Environmental Concerns							
Bottled	2.14%	4.06%	8.76%	19.87%	29.70%	19.23%	16.24%
Tap	13.33%	15.56%	11.11%	15.56%	24.44%	11.11%	8.89%
Minerals							
Bottled	1.28%	9.62%	16.24%	27.56%	20.51%	14.32%	10.47%
Tap	0.00%	11.11%	11.11%	20.00%	28.89%	17.78%	11.11%
Personal/ Family habits							
Bottled	5.13%	10.47%	11.32%	10.68%	15.38%	22.44%	24.57%
Tap	2.22%	6.67%	13.33%	6.67%	11.11%	15.56%	44.44%

Note: The most important factor is indicated by 1, and the least important factor is indicated by 7 when ranked. A *p*-value < 0.05 indicates significance. NS: not significant. Moreover, 175 students (25.4%) did not answer the rating question.

**Table 4 ijerph-21-01271-t004:** Median (IQR) of ranked factors influencing preferences for plastic bottled water (N = 468) or tap water (N = 45).

Factors	Plastic Bottled Water Median (IQR)	Tap Water Median (IQR)	Kruskal–Wallis Test *p*-Value, χ^2^ (df)
Cost	4.0 (5.0)	2.0 (4.5)	<0.001 13.97
Convenience	4.0 (4.0)	3.0 (3.5)	0.006 7.41
Safety	2.0 (2.0)	4.0 (2.0)	<0.001 35.52
Taste	3.0 (2.0)	4.0 (3.0)	0.001 10.63
Environmental Concerns	5.0 (2.0)	4.0 (3.0)	<0.001 12.53
Mineral Content	4.0 (2.0)	5.0 (2.0)	0.267
Personal/Family habits	5.0 (3.0)	6.0 (3.0)	0.028 4.84

Note: A *p*-value is for the Kruskal–Wallis test.

## Data Availability

The data presented in this study are available upon request from the corresponding author. The data are not publicly available due to confidentiality concerns.

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
