# Peer review of "Examining Drinking Water Preferences among University Students: A Comparative Assessment"

_ijerph, 2024, doi:10.3390/ijerph21101271_

Round 1

Reviewer 1 Report

Comments and Suggestions for Authors

This is a well-written manuscript in the scope of the journal, but I have some concerns about the statistics and suggestions for the structure of the manuscript.

TITLE

I suggest changing the title of the manuscript. First, change preference to choice. Second, consider using the term "comparative assessment” in the title - comparative assessment means side-by-side comparison and if you want to keep that part of the title, indicate that it is assessment among university students who chose different drinking water; or you can delete comparative

ABSTRACT

Lines 14- 16 – give the date of the study in brackets at the end of the sentence

Line 18 – Indicate the average age of the study population and the percentage of women involved in the study by number of participants

Line 22 – Environmental concerns – lower case “E”

Line 21 – 23 – In the total study population or in the 468 students who preferred plastic bottled water

STRUCTURE OF THE MANUSCRIPT

It is necessary to organize the manuscript according to the recommended template and to revise some parts. Separate the results and the discussion as stated in the recommended manuscript template on the journal's website. The discussion section can include the implications and recommendations of the study as one of the paragraphs, or the implications and recommendations of the study can run throughout the discussion. In the final paragraph of the discussion, you should emphasize the strengths and limitations of the study. The conclusion is in a separate section from the discussion, according to the journal.

INTRODUCTION

Line 122 – Why you use both examine and assess? Do you consider use of “to observe”?

MATERIALS AND METHODS

Survey questionnaire development and pilot testing – Please clarify whether the questionnaire has been validated and include reference to the validation study.

Framework – I am under the impression that the framework paragraph is part of the development of the survey, explaining what theory you used to select the possible factors that could influence the choice of drinking water. Therefore, it would be easier to understand the survey development process (factor selection) if this was explained after line 130, before the pilot test.

Variables – is also part of the draft questionnaire. The highlighting in the separate section indicates that these are all variables that are used as covariates in the statistical analysis. Therefore, combine this paragraph with the part where you talk about the development of the questionnaire

Statistical methods - have you tested the distribution of the variables? Please clarify the nature of the variables and consider accordingly whether a one-way ANOVA is appropriate or whether a non-parametric test should be used. I also suggest that you add a post-hoc analysis to confirm the differences between each group of respondents. Indicate the statistical significance level you have chosen for the analysis.

RESULTS AND DISCUSSION

Tables 1 and 3 – delete “correlation” because it was suggested that you present the results of the correlation tests and not ANOVA or chi-square

Tables 1-3 – include the p-value in the description below the table in the brackets after mentioning the test used for the analysis

Line 250 - 2600 ml daily consumption – I assume this refers to total fluid consumption, of which water is the preferred source. However, the survey only asks about water consumption so you cannot determine if this is in line with the recommendations as fluid intake can come from other sources such as tea, coffee, milk, sports drinks, food etc.

Lines 267 – 269 – explain this as differences, not correlation

Line 269 – I do not see “mineral content” in Table 2.

Table 3 – consider presenting the factors as mean ± SD or median (interquartile range) depending on the type of variable and test the differences between bottled and tap water participants. This could give a better insight into the strength of a single factor and facilitate the interpretation of the data. In addition, these parameters (%) that you have presented in Table 3 can be useful for you to deepen the discussion that you will separate from the results

Comments on the Quality of English Language The English is understandable, but there are minor errors that need to be corrected, as well as the way the results are expressed, which is emphasized in the comments.

Author Response

Dear Reviewer,

Thank you for your valuable feedback. We have carefully attended to all your comments. Please find attached the  document where the responses are in red. Also, all edits have been conducted and added are in red font in the original manuscript.

Reviewer 2 Report

Comments and Suggestions for Authors

This paper investigated drinkig water preferecnes among university students using a cross-section approach. This paper's contribution is marginal. Especially, there are several critical weakenesses.

1. In the introduction, the authors fail to address the key contribution. It should be linked to theoretical robustness. 

2. Before the materials section, the authors should establish relevent hypotheses or research questions. Withour them, why this study is essential?

3. From the academy research perspective, the use of the student sample is not adequate.

4. The authors should discuss theoretcial and practical implications based on your findings.  

Author Response

Dear Reviewer,

Thank you for your valuable feedback. We have carefully attended to all your comments. Please find attached the document with our responses in red. Also, all edits have been conducted and added are in red font in the original manuscript.

Reviewer 3 Report

Comments and Suggestions for Authors

This study deals with water consumption and preference in Qatar. the subject is very interesting. However, the manuscript has show multiple gaps that has affected its quality.

Major revision:

The main concern is related to the results presentation and analysis. In fact, the author's ideas needs to be revised and reorganized. The authors begin with a "small" description of the demographic characteristics  with multiple missed ones (presented in Table 1) (lies 201-205), They provide their preference (lines205-206) and the associated factors (without providing the description of these factors: exp: daily water consumption (line 209) ….they return to this point on line 217).

In line 223, they repeated the sentence related to the preference and in line 237 they returned to the factor "sex" that they described separately.

In line 247, they provided the results of daily consumption (that should be provided before) without any explanation and one cannot understand why the authors provided these information (in this order). (an example of disorder is just when we see that table 1 is indicated in lines 208 and 237).

The content of the titles (paragraphs) 3.5 and 3.6 is also not very clear. Try to revise

The other concern is related to the association of the discussion with the results. In fact, this method has substantially  affected he quality of the results and one cannot understand if he is reading the results or the discussion in some parts. Of notes, the discussion of the results is very simplistic and the efforts of summarizing is very limited.

The statistical analysis is another concern. After providing the revision for the asked remarks, they could complete the analysis with other tests (regression…).

Minor revision:

Delete comparative from the title

Provide the conclusion in the abstract

In line 26 you provided your recommendation and then your returned to the results (Line 27) to return later to the recommendations (without providing the conclusion). Revise

Line 133: provide the number of pre-tested questionnaire and precise that they were not included in the analysis.

Line 206-207: reformulate the sentence (exp: followed by those having no preference..).

Line 209: daily water consumption : se the comment above   

You should provide the tables just after the corresponding results

Line 262: there is no question related to water (consumption) knowledge here. The term knowledge should be deleted.

Line 269: "….was seen with mineral content…". This may be "…was seen with the perception regarding mineral content". Revise at your convenience.

Add the limitations of the study at the last of the discussion

Add a conclusion to the study (you provided recommendations in lines 379-383 not a conclusion).

At last, you should provide more information about the questions used in the questionnaire (in the methods or as a supplementary material)

Comments on the Quality of English Language

Minor editing required

Author Response

Dear Reviewer,

Thank you for your valuable feedback. We have carefully attended to all your comments. Please find attached document with our responses in red. Also, all edits have been conducted and added are in red font in the original manuscript.

Round 2

Reviewer 2 Report

Comments and Suggestions for Authors

The revision is significant.

Well done.

Author Response

We would like to extend our sincere appreciation for your comments and feedback. Your insights have significantly improved the quality of our manuscript.

Reviewer 3 Report

Comments and Suggestions for Authors

I would thank the author for their efforts to improve the quality of the manuscript. However, the manuscript is still lacking a certain rigor and more efforts should be done, especially in the results presentation and the discussion.

Major revisions:

You should use subtitles to separate the demographic characteristics from water consumption (quantity and preference) and make each table after the corresponding results (you should separate table 1 into 2 tables).

In addition, the description of lines 254–278 is not in relation to the content of table 1. Where can we find these results ? you should provide a table containing all these results and especially the results of the statistical analysis (p value) you provided.

The same remark applies for line 289-299 while table 2 is not indicated in the text. The same remark also for the following table which should be placed just after their corresponding results.

You should provide the results of statistical analysis in table 2

Lines 314-323: where can we find these results?

How did you calculate the median in the ranking factor provided in table 4? (it is not shown in the methods).

Discussion:

The discussion provided in line 385-401 has no relation with the subtitle (female preference). Also the ideas are not well organized, you begin with water consumption in the results but this point was treated in the end of the discussion.

I suggest deleting subtitles from the discussion.

I have also other comments:

Reformulate the sentence of lines 19-20 (try to avoid the repetitions).

Lines 191-200: try to reformulate to avoid the repetition of the word "categorical".

Line 245: "the response rate was n=688…." . Revise the response rate may be a %).

Line 294: "….significant association between (χ(6) = 18.07, p = 0.006, Cramer’s V = 0.188) ),". Between what?

Table 3: what did the bold character mean?

Line 334: p value should be in lower case.

Comments on the Quality of English Language

Moderate English editing required

Author Response

Dear Reviewer,

Thank you for taking the time to thoroughly review our manuscript. We appreciate your observant and detailed feedback and suggestions that have greatly strengthened our work. We have addressed all your comments. The document is attached. Additionally, edits and additions have been made in blue font in the original manuscript.
